# Disruption of *LLM9428*/*OsCATC* Represses Starch Metabolism and Confers Enhanced Blast Resistance in Rice

**DOI:** 10.3390/ijms23073827

**Published:** 2022-03-30

**Authors:** Yongxiang Liao, Asif Ali, Zhenzhen Xue, Xia Zhou, Wenwei Ye, Daiming Guo, Yingxiu Liao, Pengfei Jiang, Tingkai Wu, Hongyu Zhang, Peizhou Xu, Xiaoqiong Chen, Hao Zhou, Yutong Liu, Wenming Wang, Xianjun Wu

**Affiliations:** State Key Laboratory of Crop Gene Exploration and Utilization in Southwest China, Rice Research Institute, Sichuan Agricultural University, Chengdu 611130, China; liaoyongxiang123@163.com (Y.L.); asifali@sicau.edu.cn (A.A.); xuezhenzhen202202@163.com (Z.X.); h1056567243@126.com (X.Z.); yewenwei0723@163.com (W.Y.); daiming_guo@outlook.com (D.G.); liaoyingxiu827@163.com (Y.L.); jiangpengfei1997@outlook.com (P.J.); wtksicau@163.com (T.W.); zhanghysd@163.com (H.Z.); xpzhxj@163.com (P.X.); xiaochenq777@126.com (X.C.); zhouhao666@foxmail.com (H.Z.); liuyutong910617@163.com (Y.L.)

**Keywords:** catalases, lesion mimic mutant, blast resistance, starch granule

## Abstract

Catalases (CATs) are important self-originating enzymes and are involved in many of the biological functions of plants. Multiple forms of CATs suggest their versatile role in lesion mimic mutants (LMMs), H_2_O_2_ homeostasis and abiotic and biotic stress tolerance. In the current study, we identified a *large lesion mimic mutant9428* (*llm9428*) from Ethyl-methane-sulfonate (EMS) mutagenized population. The *llm9428* showed a typical phenotype of LMMs including decreased agronomic yield traits. The histochemical assays showed decreased cell viability and increased reactive oxygen species (ROS) in the leaves of *llm9428* compared to its wild type (WT). The *llm9428* showed enhanced blast disease resistance and increased relative expression of pathogenesis-related (PR) genes. Studies of the sub-cellular structure of the leaf and quantification of starch contents revealed a significant decrease in starch granule formation in *llm9428*. Genetic analysis revealed a single nucleotide change (C > T) that altered an amino acid (Ala > Val) in the candidate gene (*Os03g0131200*) encoding a CATALASE C in *llm9428*. CRISPR-Cas9 targetted knockout lines of *LLM9428/OsCATC* showed the phenotype of LMMs and reduced starch metabolism. Taken together, the current study results revealed a novel role of *OsCATC* in starch metabolism in addition to validating previously studied functions of CATs.

## 1. Introduction

CATALASE (CAT) is an antioxidant enzyme located in the cytoplasm, mitochondria and chloroplast. It mainly decomposes H_2_O_2_ produced in the mitochondrial electron transport chain and photorespiration and prevents plant cells from oxidative stress damage [1]. Previous studies have demonstrated that CAT plays an important role in plant growth and response to biotic and abiotic stresses [2]. At present, three types of CAT genes e.g., *OsCATA*, *OsCATB* and *OsCATC* have been characterized in rice. *OsCATA* is expressed in different tissue including roots, leaves and stem, where the highest expression level was observed in leaves [1,3]. Under a copper stressed environment, the expression level of *OsCATA* was down-regulated in germinating seeds and resulted in an increase of H_2_O_2_ [4]. The expression level of *OsCATA* and *OsCATC* were repressed under a water stress environment, but the expression of *OsCATB* was up-regulated and prevented the excessive accumulation of H_2_O_2_ [4]. The Llss of function mutant of *OsCATC* revealed increased H_2_O_2_ content that activates nitrate reductase and promotes the production of nitric oxide (NO), which have been reported to be involved in H_2_O_2_-induced cell death in rice [5]. *OsCATC* also has been reported to play its role in the oxidation of glycolate into CO_2_ and increases nitrogen content, photosynthetic efficiency and grain yield [6].

A leaf is the main source organ of photosynthesis in plants. Photosynthetic products are stored in leaf chloroplasts in the form of excessive starch during the daytime. At night, starch is hydrolyzed into sucrose with the help of different enzymes to sustain sugar metabolism [5,7]. Like H_2_O_2_, the excessive accumulation of starch in leaves also plays a role in programmed cell death (PCD) and causes leaf senescence by damaging the thylakoid membrane [8]. At present, starch metabolic pathway in *Arabidopsis* is clear, however, there are relatively few studies in rice, e.g., *TREHALOSE-6-PHOSPHATE PHOSPHATASE 1 (**OsTPP1)*, *SUCROSE PHOSPHATE SYNTHASE 1 (**SPS1)*, *ISOAMYLASE 3 (OsISA3)* [9,10,11] and *GLUCAN WATER-DIKINASE 1 (OsGWD1)* [12]. *OsGWD1* encodes a key enzyme for starch degradation and its loss of function mutant produced excessive starch in leaves [12]. Overexpression of *OsTPP1* enhances stress tolerance to salt and cold. Mutations in starch-producing genes can cause abnormalities in starch metabolism and activate the expression of stress-responsive genes [9]. Similarly, in rice, *SEKIGUCHI LESION (OsSL)* encodes a *CYTOCHROME P450 MONOOXYGENASE* located in the endoplasmic reticulum and its loss of function mutant *early lesion leaf1* (*ell1)* showed lesions like phenotype and enrichment of starch particles [2]. Previous studies have highlighted the role of starch in the development of leaf lesions, however, the relationship between starch metabolism and PCD remained largely unknown.

Lesion-mimic mutants (LMMs) are present in hypersensitive (HR)-like lesions without pathogenic attack, and are used as an excellent source for the exploration of PCD and its molecular mechanism in plants [13]. Several LMMs have been characterized in *Arabidopsis*, barley (*Hordeum vulgare*), maize (*Zea mays*) and rice [14,15,16,17]. In rice, more than 20 LMM-related genes have been cloned and characterized, which encode different types of proteins e.g., AAA-type ATPase [18,19], *CYTOCHROME P450 MONO-OXYGENASE* [20], *HEAT STRESS TRANSCRIPTION FACTOR PROTEIN* [21], *ATP-CITRATE LYASES* [17], *CULLIN 3-BASED RING E3 UBIQUITIN LIGASES* [22] and *XANTOXIN DEHYDROGENASE* [23]. Therefore, the genetic regulation of lesion mimic formation in plants is controlled by multistep and complex processes. Despite a variety of LMMs having been isolated and characterized, the relationship between starch metabolism and lesion formation is elusive.

In the current study, a *large lesion mimic mutant 9428* (*llm9428*) was screened from an Ethyl methanesulfonate (EMS) mutagenized population of an *indica* cv. Yixiang 1B. The mutant (*llm9428)* displayed typical phenotypes of an auto-immune response, such as spontaneous cell death, up-regulation of PR-genes, enhanced blast disease resistance and reduced starch metabolism. Loss of function mutant and CRISPR/Cas9 knock-out lines confirmed the role of *OsCATC/LLM9428* in the immune response, retarded plant growth and development, and repression of starch metabolism. These results provide further insights into the molecular function of *OsCATC* in lesion mimic formation, disease resistance and starch metabolism in rice.

## 2. Results

### 2.1. llm9428 Showed a Stable Phenotype of Lesion Mimic Mutant

To understand the genetic and molecular basis of the lesion mimic phenotype, we observed the growth pattern of *llm9428* and compared it to a wild-type (WT). At the seedling stage, the WT showed normal phenotypic traits, however, *llm9428* showed stunted and retarded growth (Figure 1A). Large lesion-like spots began to appear in the leaves and expanded to the whole leaf with further growth and development in *llm9428* (Figure 1B). Various agronomic traits, e.g., plant height, tiller number per plant, the number of grains per panicle, 1000-grains weight and seed-setting rate were significantly reduced in *llm9428* (Figure 1C–G). Phenotypic and agronomic traits analyses revealed that *llm9428* showed lesion mimic phenotypes on its leaves coupled with a significant decrease in yield-related traits.

### 2.2. llm9428 Showed Decreased Cell Viability and Increased (Reactive Oxygen Species) ROS

To assess the changes associated with the lesion mimic phenotype on leaves, trypan blue staining was used to check cell viability in *llm9428* and the WT (Figure 2). Trypan blue solution produced light stains in the WT and dark and larger stains on leaves of *llm9428*, which indicated the cell viability in *llm9428* had been compromised (Figure 2A). Excessive ROS accumulation can be a result of genetic factors and environmental stresses and cause damage to the cellular machinery, proteins, DNA and structure of macromolecules [24]. Additionally, 3,3′-diaminobenzidine (DAB) staining revealed high levels of H_2_O_2_ in *llm9428* as compared to the WT (Figure 2B). These histochemical assays revealed that the lesion mimic phenotype was associated with increased cell death and ROS.

### 2.3. llm9428 Showed Enhanced Blast Disease Resistance

Recent studies have revealed that cell death in blast disease is often associated with the immune response [25]. To check the immune response, leaves of the WT and *llm9428* were inoculated with a *Magnaporthe oryzae* strain Zhong10-8-14 (GZ8) at the four-leaf stage. Although, initially both the WT and *llm9428* showed affinity to lesion development, however, after six days of inoculation the lesions were developed more vigorously in the WT than *llm9428* (Figure 3A). The length and width of the lesions were also significantly increased in the WT compared to *llm9428* (Figure 3B,C). Previous studies have reported increased expression of PR-genes in mutants with enhanced blast resistance [26]. Accordingly, the relative expression of *PR* genes e.g., *PR1a*, *PR1b* and *PR10* was higher in *llm9428* than the WT (Figure 3D). These results support that *llm9428* has enhanced blast disease resistance.

### 2.4. Suppression in the Development of Chloroplast and Starch Metabolism in llm9428

Previous studies have indicated the role of ROS in biotic and abiotic stress including damage to leaf structures especially chloroplast [24,27]. To assess whether the ROS associated with lesions caused any damage to the subcellular structure of *llm9428,* we analyzed the chloroplast and mesophyll cell morphology using electron micrographs of the WT and *llm9428* at the flowering stage (Figure 4). The WT revealed a well-developed membrane system, starch granules and grana (Figure 4A,B). However, in *llm9428,* the grana stacks were found degenerated and less dense with a significant decrease in the area of starch granules (Figure 4C,D). The thylakoid membrane system and membrane spacing were also found significantly disturbed compared to the WT. These results indicated the development of chloroplast and starch granule formation was suppressed in *llm9428*.

### 2.5. Reduction in Starch Content and Expression of Its Related Genes in llm9428

To answer whether reduced starch granules in *llm9428* have caused any change in starch contents, the starch contents were quantified from the leaves and grains of the WT and *llm9428* (Figure 5). Results showed that total starch content in the leaves of *llm9428* was significantly lower than the WT (Figure 5A). Starch metabolism was reported to be changed according to the day length, temperature and circadian clocks [28]. The total starch contents “before end of day” and “before end of night” were quantified, and both were significantly decreased in *llm9428* (Figure 5A). Reduction of the starch granules area and number were also reported to distress the starch signalling pathway by affecting the relative expression of starch-synthesis genes [29,30]. Relative expression of different structural components of starch, transport and its synthesis genes were determined in the WT and *llm9428* (Figure 5C). Among them, the relative expression of *STARCH SYNTHASE I*
*(OsSSI)*, *OsSSIIb*, *ADP-GLUCOSE PYROPHOSPHORYLASE LARGE SUBUNIT 1 (O**sAGPL1)* and *OsAGPL3* especially starch-synthesis and structural component genes were significantly lower in *llm9428* compared to the WT. These results indicated that the reduced accumulation of starch granules was attributed to the relative expression of starch genes in *llm9428*.

### 2.6. Genetic Analysis of llm9428

To find whether the phenotype of *llm9428* is controlled by a dominant or recessive gene, *llm9428* was crossed as a female and male parent to Yixiang 1B to develop two populations (*llm9428* × Yixiang1B and Yixiang1B × *llm9428*). None among the F_1_ plants of both populations showed any phenotype of LMMs. In F_2,_ 132 among 584 plants from *llm9428* × Yixiang1B population, and 219 among 890 plants from Yixiang1B × *llm9428* population showed the phenotype of LMMs (Table 1). Chi-square analysis of observed trait and segregation ratios (3:1) revealed that the phenotype of LMM in *llm9428* is controlled by a single recessive nuclear gene.

To map a locus controlling the phenotype of LMMs, *llm9428* was crossed with a japonica cv. *02428* to develop a mapping population (*llm9428* × *02428*). 512 pairs of simple sequence repeats (SSR) markers, evenly distributed on all 12 chromosomes of rice, were applied to the F_2_ individual showing the phenotype of LMMs. The presence of polymorphic bands between dominant (F_2_ plants without LMM phenotype) and recessive (F_2_ plants with LMM phenotype) plants showed that the locus is present on chromosome 3 between SSR markers RM5474 and Os.3.6.3 (Figure 6A and Appendix A). To further narrow down, the polymorphic markers were searched from the primary mapped region. Screening of 276 plants from the mapping population revealed that the locus can be further narrowed down between SSR markers RM14360 and Indel-304 (Figure 6B). Screening of a further 1806 plants with newly developed indel markers revealed the mutation in *llm9428* was located between the indel markers Indel-312 and Indel-319. A list of primers used in the SSR and indel markers analysis is given in Appendix A (Figure 6C and Appendix A). Genetic linkage map, SSR and indels markers revealed that the candidate gene locus is positioned between Indel-312 and Indel-319 within a physical distance of 59 kb (Appendix A). According to the Rice Genome annotation project accessed on 15 June 2021 (http://rice.plantbiology.msu.edu/), this region contains a gene “*Os03g0131200*”, which encodes a CATALASE C (OsCATC). To further verify the mutation, we deep sequenced the *OsCATC* gene from the WT and *llm9428*. Comparative sequence analysis revealed a one bp mutation (C > T) of 752^nd^ nucleotide in the third exon of *OsCATC* in *llm9428*. This is a missense mutation and caused the change of amino acid alanine (Ala) to valine (Val) (Figure 6D). The chromatogram of *llm9428* validated the presence of a single nucleotide polymorphism (SNP) in *OsCATC* (Figure 6E). Hence, a tentative name “*LLM9428*” was assigned to the candidate gene of *llm9428*.

### 2.7. CRISPR-Cas9 Targetted Knockout Lines of LLM9428/OsCATC Showed the Phenotype of LMMs and Reduced Starch Metabolism

To confirm whether phenotypes of *llm9428* were associated with the mutation of *LLM942,* we developed knockout (KO) lines of its candidate gene. Three independent KO lines were developed by targetting the first exon of *LLM9428* (Figure 7A). Sequencing revealed the successful deletion of 4 bp, 2 bp and 1 bp deletion in *ko1*, *ko2* and *ko3,* respectively (Figure 7A and Appendix A). Transgenic lines of *LLM9428* displayed lesion mimic spots on their leaves (Figure 7B). *LLM9428* encodes a CATALASE C (OsCATC), an enzyme involved in the hydrolysis of H_2_O_2_, and its activity was significantly decreased in *ko-1* compared to Nipponbare (NIP) (Figure 7C). A previous study has reported the role of CATALASE in the regulation of intracellular levels of H_2_O_2_ [31]. Accordingly, the quantification assays indicated that the CAT activity was decreased (Figure 7C) and H_2_O_2_ contents were significantly increased in *ko1* compared to NIP (Figure 7D).

Further, to test whether the deletion in ko1 has caused any changes in granular starch, as previously observed in *llm9428*. Leaf sub-cellular structure of *ko-1* at “before end of day” and “before end of night” were examined using transmission electron micrography at tillering stage (Figure 7E–H). The results indicated that the mesophyll cells and chloroplast in *ko1* had degenerated and the number of starch granules was found to be significantly decreased compared to NIP. In addition, other phenotypic and agronomic traits e.g., plant height, tiller number, number of grains per panicle, 1000-grains weight and seed setting rate were significantly decreased in *ko1* and comparable to *llm9428* (Figure 7I–M). Taken together, these results not only validate the role of *LLM9428* in lesion mimic formation but also present its novel role in starch granule formation.

## 3. Discussion

CATs are self-originating enzymes and have been reported to play their role in ROS homeostasis, nitrogen use efficiency, leaf senescence, cell death, photorespiration and conferring blast disease resistance [4,5,6,31,32]. The diversity in the role of these enzymes indicates their importance in plant function and metabolism. Versatility in the functions of these enzymes is attributed to their role in H_2_O_2_, which acts as a modulator of oxidative stress under biotic and abiotic stresses [33]. In the current study, a novel role of *OsCATC* in regulating starch granule formation in *llm9428* was reported. The *llm9428* showed a typical phenotype of LMMs and showed brown color lesions on the surface of leaves. These lesions are necrotic spots and are reported to vary in different mutants in terms of size and color e.g., reddish-brown [23], orange [24], yellow-brown [25], brown [26] and yellow-green striped necrotic spots [27]. These spots also vary in terms of shape and range from a diameter of about 1 mm in *spl3* [28] to more than 10 mm in *spl5* [29] and *noe1* [17]. However, *llm9428* showed relatively larger white and green lesion-like spots, which usually start from the lower leaves and gradually spread to all the leaves. The *llm9428* displayed decreased height, reduced tillering, reduced 1000-grains weight (Figure 1), and enhanced resistance to rice blast (Figure 3). Typical phenotypic traits of *llm9428* are consistent with the previous findings of LMMs [23,34,35].

Starch granules in leaf chloroplasts are a temporary storage form of photosynthetic products. These are synthesized in amyloplasts and stored in the endosperm [30,31]. Dysregulation of starch metabolism in leaves will lead to the excessive accumulation of starch granules in leaves [32]. Starch granule formation was affected due to the loss of function of *ISOAMYLASE* in barley [36]. However, in the current study, degenerated starch granules were found in chloroplasts of *llm9428* due to loss of function of *OsCATC* (Figure 4). Total starch contents in leaves and grains of *llm9428* were significantly decreased compared to the WT (Figure 5). Moreover, the decrease in the relative expression of starch synthesis and transport-related genes highlights the role of these enzymes in starch granule formation. It can be speculated that the obstruction of photosynthetic activity may have caused changes in starch granules in *llm9428*. A significant decrease in the starch accumulation in the leaves of *llm9428* revealed an opposite trend to the leaf pre-senescence mutants *starch accumulating 4 (ossac4)* [33]. A significant decrease in the starch content in *llm9428* could be due to decreased photosynthesis, which was affected by lesions. Moreover, the reduced starch content in leaves (Figure 5A,B) and the large area of disease-like spots refer to the decreased photosynthetic rate in *llm9428*.

CAT widely exists in animals, plants and microorganisms, and its main function is to decompose H_2_O_2_ into oxygen and water to maintain the balance of ROS in the cell. In the current study, the CAT activity was significantly decreased but on the contrary, the H_2_O_2_ contents were increased in *llm9428* compared to the WT (Figure 7). H_2_O_2_ is a stable reactive oxygen species that plays an important regulatory role in biological processes such as plant growth and development, senescence, biotic stress and abiotic stress [35,36,37]. In addition, the expression levels of PR genes e.g., *PR1a*, *PR1b* and *PR10* were higher in *llm9428* than the WT and showed enhanced resistance to *M*. *oryzae* (Figure 3). Therefore, results revealed that mutation in *LLM9428* reduces CAT activity, promotes H_2_O_2_ content, which increases cell death and triggers an immune response. Similar findings have already been observed in LLMs e.g., *spl5* [29], *spl3* [28] and *lmm9150* [11]. Additionally, *llm9428* will serve as a source to study the effect of lesions on starch granule formation. The relationship between H_2_O_2_ and starch metabolism in leaf chloroplasts is still elusive and deserves further study.

Fine-mapping of *LLM9428* between the molecular markers of Indel-312 and Indel-319 on chromosome 3 was mapped in a physical interval of about 59 kb. This interval contains other eight annotated genes, among them *OsCATC/NOE1* catalyzes the hydrolysis of H_2_O_2_. Lin et al. already reported the involvement of H_2_O_2_ in inducing leaf cell death by promoting the production of nitric oxide that results in the appearance of massive lesions on the leaves of the *noe1* at the seedling stage [17]. Whether other genes in this region have a linkage with *OsCATC/NOE1,* and how the final phenotype of *llm9428* is controlled at the molecular and proteomic level should be studied in the future. The current study adds a novel role in the multiple biological functions of *OsCATC*.

## 4. Material and Methods

### 4.1. Plant Materials

The seeds of a maintainer line (Yixiang 1B), the backbone of more than 50 hybrids and bred from the Institute of Agricultural Sciences, Yibin, Sichuan, China, were soaked (18 h) in a 1.2% solution of EMS for mutagenesis. The seeds were thoroughly washed in running water for 72 h and M_0_ population was grown in the field for phenotypic screening. A disease-like phenotype showing mutant *llm9428* was screened from the mutagenized population and Yixiang 1B was taken as the Wild Type (WT) throughout the study. The phenotypic trait of LMM was stable and inherited in all succeeding segregating generations of the mutagenized population. The mutant was used alternatively as a male and a female parent and crossed with Yixiang 1B to develop two populations to study the segregation behavior of the candidate gene (Table 1). At the same time, *llm9428* was crossed with a *japonica* cv. 02428 to generate a mapping population, which was used for gene mapping and SSR analysis. The F_2_ generation of the mapping population was used for genetic analysis. All plants were grown under natural field conditions in an experimental area of Rice Research Institute, Sichuan Agricultural University, Chengdu (N30.67°, E104.06°), China.

### 4.2. Investigation of Agronomic Traits

Data of agronomic traits were taken as an average number of 10 randomly selected plants of WT and *llm9428* at the maturity stage.

### 4.3. Trypan Blue Staining

Trypan blue staining was used to identify dead cells. Samples were taken from the three-leaf stage, and immediately placed into trypan blue solution (ddH_2_O: lactic acid: glycerol: phenol: ethanol in the volume of 50 mL: 50 mL: 50 mL: 50 mL: 300 mL, respectively). The solution containing samples was boiled for five minutes and thoroughly rinsed with sterilized water. Then samples were placed in a decolorization solution to decolorize the dying. The decolorized green leaves were placed in glycerol (50%) for long-term storage and leaf sections were observed and photographed.

### 4.4. DAB Staining

DAB staining was used to detect the ROS from the samples. The selected samples were soaked in a DAB solution (1 mg/mL) and the pH of the solution was adjusted to 3.8 with concentrated HCl and stored in the dark at 4 °C. The solution containing samples was incubated (25–29 °C) overnight for staining. Samples were decolorized with a solution (75% alcohol, 5% glycerol mixture). After decolorization, samples were directly observed and photographed.

### 4.5. Identification of Rice Blast and Pathogen Inoculation

*M. oryzae* strain Zhong10-8-14 (GZ8) was cultured on sterilized plates containing oatmeal-tomato-agar (OTA) medium for 10 days at 28 °C with 12 h light and 12 h dark cycles, as previously reported by Chen et al. [37]. All-grown hyphae were scraped off with distilled water and to promote sporulation plates were further incubated for four days under the above-mentioned conditions. Fresh leaves, collected from the four-leaf stage were used for punch inoculation. The concentration of spores was adjusted to 5 × 10^5^ conidia mL^−1^ for inoculation. 10 μL spore suspension was applied to each wounded site of the leaves. The inoculated leaves were incubated on 0.1 mmol/L 6-benzyl-adenine buffer. After 6 days, leaves were observed for blast lesions and size measurements.

### 4.6. Quantitative Relative Expression of Genes

RNA was extracted using Trizol reagent (purchased from Invitrogen) from rice leaves. High-quality RNA was reverse transcribed into cDNA with Takara’s PrimeScript^TM^ RT Kit with a gDNA eraser (Takara, Dalian, China) according to the manufacturer’s instruction. Takara SYBR^®^ PrimeScript™ RT-PCR kit (Takara, Dalian, China) was used to determine the expression of the particular genes. Quantitative expression analysis was carried out by the software Bio-Rad CFX Manager V2.0 software installed in the qPCR instrument. Actin was used as an internal control. The quantitative expression data were analyzed according to 2 ^ΔΔ^CT method proposed by Livak et al. [38]. All primer sequences used for qPCR are listed in Appendix A.

### 4.7. Transmission Electron Microscopic Observations

At tillering stage, the leaves with particular disease-like spots were selected from *llm9428* and at the same stage from the WT. The samples were pre-fixed in a solution (3% glutaraldehyde, 1% osmium tetroxide) and then fixed in acetone. Subsequently, the dehydration gradients (30%, 50%, 70%, 80%, 90%, 95% and 100%) were used to wash samples. The dehydrated cells were successively passed through a dehydrating agent and embedded in epoxy resin (Sigma–Aldrich Epon 812, Fluke, Beijing, China). The infiltrated samples were put into an appropriate mold and polymerized to form a solid matrix (also called an embedding block). The samples were cut into semi-thin (500–800 Ǻ) sections with an ultramicrotome (Leica EM UC7, Leica Microsystems, Wetzlar, Germany). After cutting, samples were double-stained with uranyl-acetate and lead citrate and washed with deionized water before observation under the transmission electron microscope (HITACHI HT7700, Tokyo, Japan).

### 4.8. Determination of Starch Content

At the tillering stage, the samples were harvested and leaves were ground into powder form. The mature grains were pulverized into flour form. The starch contents were estimated with a starch detection kit (YX-C-C400) purchased from SINOBESTBIO, according to the manufacturer’s instructions. The same quantity (0.1 g) from the WT and *llm9428* samples were used to measure the starch contents and data was recorded from three individual biological repeats.

### 4.9. Gene Mapping and Genetic Analysis

Map-based cloning and SSR marker analyses were performed according to the study of Ali et al. [39]. Individuals of F_2_ populations (*llm9428* × Yixiang1B and Yixiang1B × *llm9428*) were used for inheritance behavior (dominance or recessiveness) of the *llm9428* phenotype. The *llm9428* was crossed with a japonica cv. 02428, and F_2_ individuals of mapping population (*llm9428* × 02428) were used for SSR marker analysis and fine mapping. The primers used in primary and fine mapping are listed in Appendix A. The genetic linkage map of chromosome 3 and physical distance are given in Appendix A. The mutation was confirmed by individual primer amplification followed by deep sequencing from the WT and *llm9428*.

### 4.10. Construction of Knock-Out Vector and Genetic Transformation

The construction of the CRISPR/Cas9 knockout (KO) vector was performed according to the previous study of Xie et al. [40]. The vector used in the CRISPR/Cas9 system was provided by Professor Liu Yaoguang of South China Agricultural University. The specific construction steps were as follows, two targets, sequence 1 (CCTTCTGGAGGACTACCACCTGG) and sequence 2 (GAAGCTGGCCAACTTCGACAGGG) were selected according to the website accessed on 12 April 2021 (http://skl.scau.edu.cn/) developed by Professor Liu Yaoguang’s research group and oligos were designed for knockout of the target sequence. The possible off-target effects were prevented using the blast search of the target sequence in the Gramene and NCBI database. The KO constructs were developed according to instructions provided with the Kit (BGK032) purchased from Hangzhou Biogle Biotechnology Co., Ltd. The target single-guide RNA (sgRNA) was constructed into the expression cassette of the pYLCRISPR/Cas9P35S-H vector and digested with BsaI. The recombinant vector was transformed into *E.coli* strain Trans-DH5α according to the guidelines of Ma et al. [41]. Positive clones were confirmed using sequencing of individual clones and positive vectors were sent to Boyun Biotechnology Co., Ltd. for transforming into calli of *japonica* cv. Nipponbare according to guidelines of Toki et al. [42]. The homozygous T_3_ plants were used for phenotypic analysis.

## 5. Conclusions

In the current study, *llm9428* mutant displayed the typical phenotype of LMMs e.g., spontaneous cell death and enhanced rice blast resistance. So far, previous studies did not report the effect of LLMs on starch metabolism. Map-based cloning revealed that the *LLM9428*, encoding a *CATALASE C* played a role in repressing the metabolism of starch granule formation in *llm9428*. The knockout lines of *LLM9428* also displayed phenotypes of lesion mimic, reduced starch metabolism, and retarded plant growth and development, which were comparable to that of *llm9428*. Together, our data demonstrated the novel functions of *LLM9428*/*OsCATC* in starch metabolism, in addition to its function in plant growth and disease resistance.

## Figures and Tables

**Figure 1 ijms-23-03827-f001:**
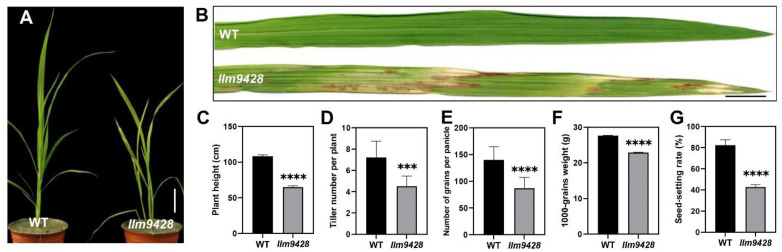
Phenotypic observation of wild type and *llm9428*. (**A**) Plant of WT and *llm9428* at the seedling stage, (**B**) The phenotype of WT leaf and *llm9428* leaf at the seedling stage. Comparison of plant height (**C**) tiller number per plant (**D**) number of grains per panicle (**E**) 1000-grains weight (**F**) and seed setting rate (**G**) between WT and *llm9428* respectively. Statistical significance was analyzed using Student’s *t*-test, *** and **** indicate *p* < 0.001 and *p* < 0.0001, respectively. Scale bar in (**A**,**B**) = 10 cm. Data presented in (**C**–**G**) are the average of *n* = 10 plants.

**Figure 2 ijms-23-03827-f002:**
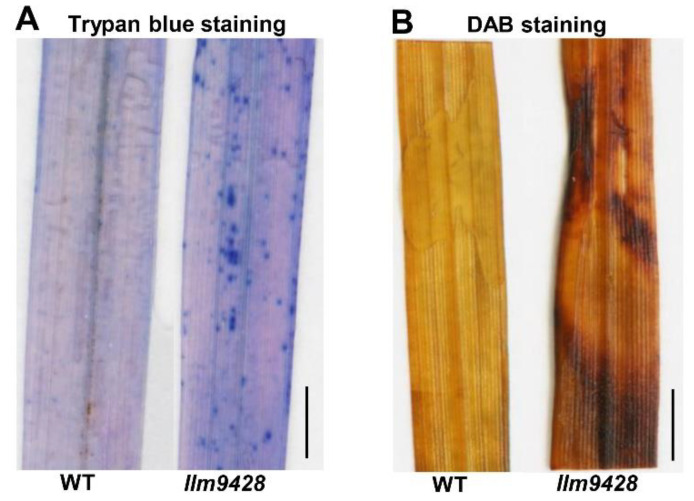
Histochemical staining of WT and *llm9428*. (**A**) Trypan blue staining of leaf section of WT and *llm9428*. (**B**) DAB staining of leaf section of WT and *llm9428*. Scale bar: 1 cm.

**Figure 3 ijms-23-03827-f003:**
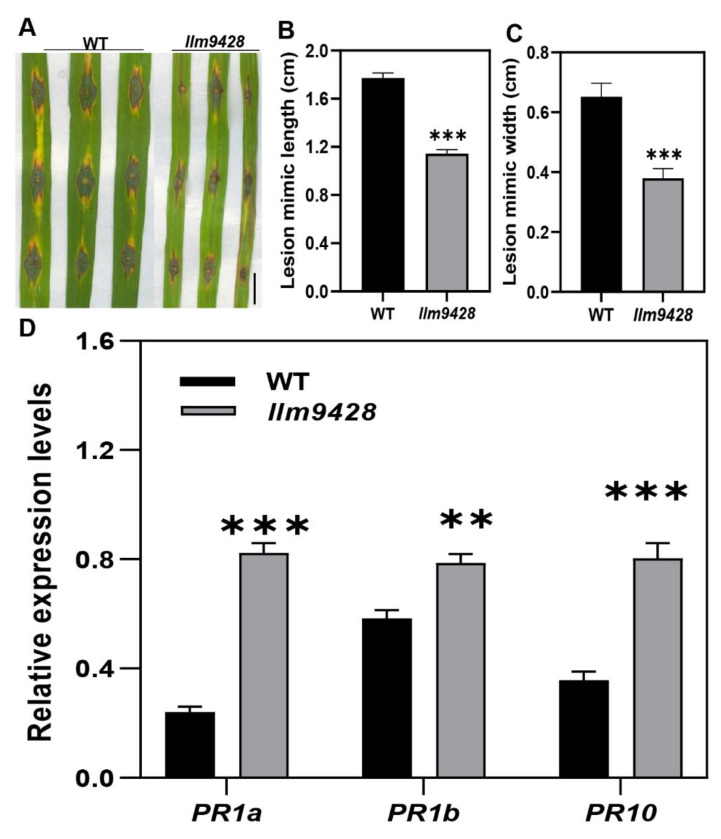
*llm9428* has enhanced blast disease resistance compared to WT. (**A**) The phenotype of disease lesions of WT and *llm9428* at four-leaf stage, (**B**) length and (**C**) width of disease lesions in WT and *llm9428*. (**D**) The relative expression levels of PR-genes in WT and *llm9428*. The data presented in B-C is the average of 10 leaves. Mean and SD were obtained from three independent measurements. Statistical analysis was performed using Student’s *t*-test, ** and *** indicate *p* < 0.01 and *p* < 0.0001, respectively. Scale bar: 1 cm in (**A**).

**Figure 4 ijms-23-03827-f004:**
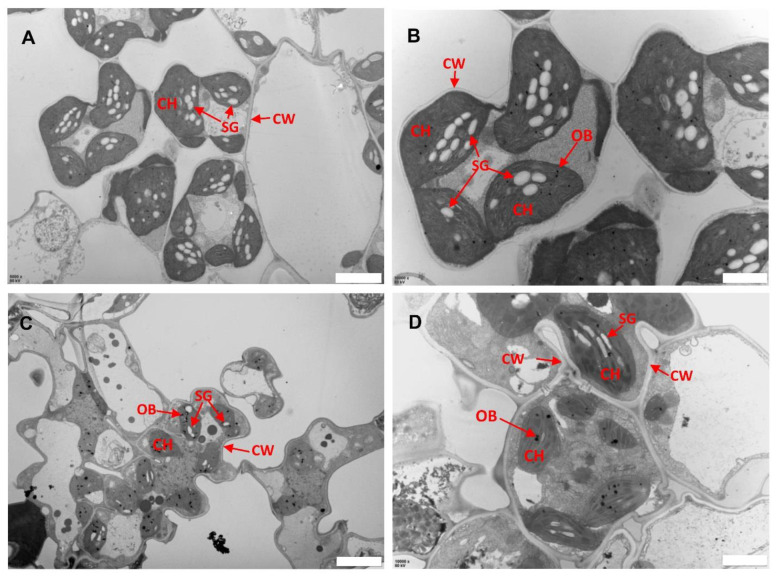
Transmission electron micrographs of leaf sub-cellular structure of WT and *llm9428*. (**A**,**B**) Subcellular structure of leaf chloroplast of WT, and *llm9428* (**C**,**D**) at flowering stage under a transmission electron microscope. Where CW: cell wall, CH: chloroplast; SG: starch granules and OB: osmiophilic body. Scale bar: 2 μm in (**A**,**C**) and 5 μm in (**B**,**D**).

**Figure 5 ijms-23-03827-f005:**
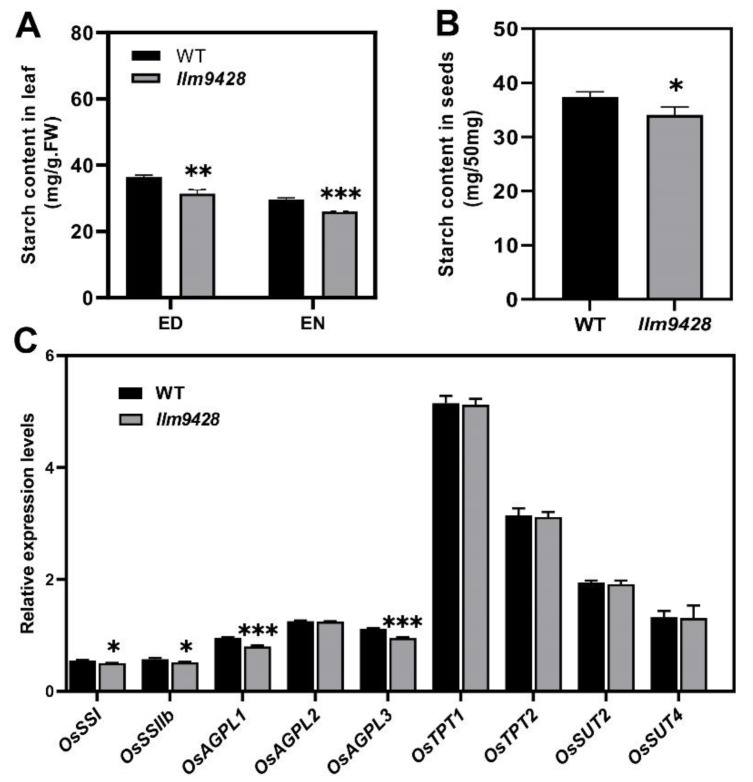
Comparison of total starch contents and relative expressions of its related genes in WT and *llm9428*. (**A**) Comparison of starch contents in the second leaves between WT and *llm9428* mutant at tillering stage (ED: end of day, EN: end of night). (**B**) Comparison of starch contents in mature grains between WT and *llm9428*. (**C**) Relative expression levels of starch metabolism-related genes in leaves of WT and *llm9428* at heading stage. Statistical analysis was performed using Student’s *t*-test, *, ** and *** indicate *p* < 0.05, *p* < 0.01 and *p* < 0.001, respectively. Mean and SD were obtained from three individual measurements.

**Figure 6 ijms-23-03827-f006:**
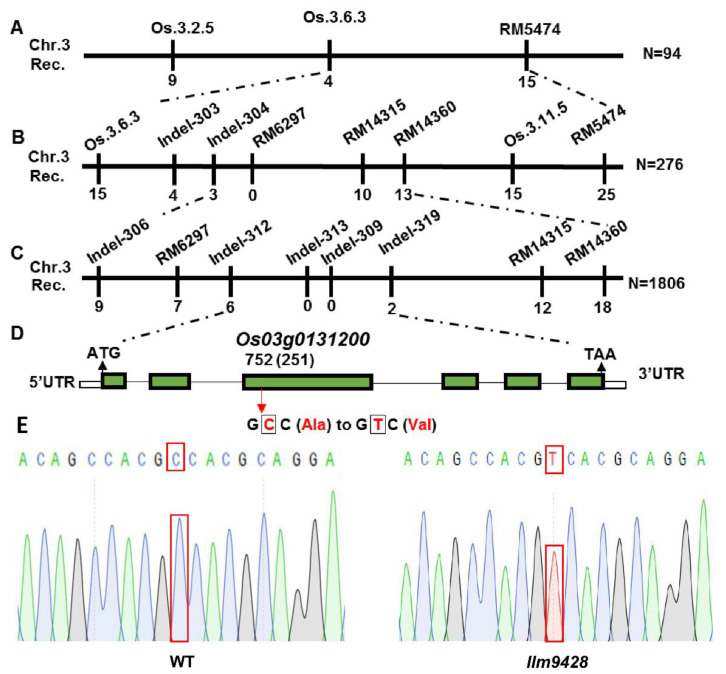
Gene mapping of candidate gene of *llm9428*.(**A**) The position of the candidate gene of *llm9428* on chromosome 3 between SSR marker Os.3.6.3 and RM5474, (**B**) The further narrowed down region of the candidate gene of *llm9428* between Indel marker RM14360 and Indel-304, (**C**) The further narrowed down region of the candidate gene of *llm9428* between Indel marker Indel-312 and Indel-319, (**D**) The structure of the candidate gene, which has six exons and five introns are indicated by green rectangles and black lines respectively. (**E**) Chromatograms showing the sequence comparison of WT and *llm9428*.

**Figure 7 ijms-23-03827-f007:**
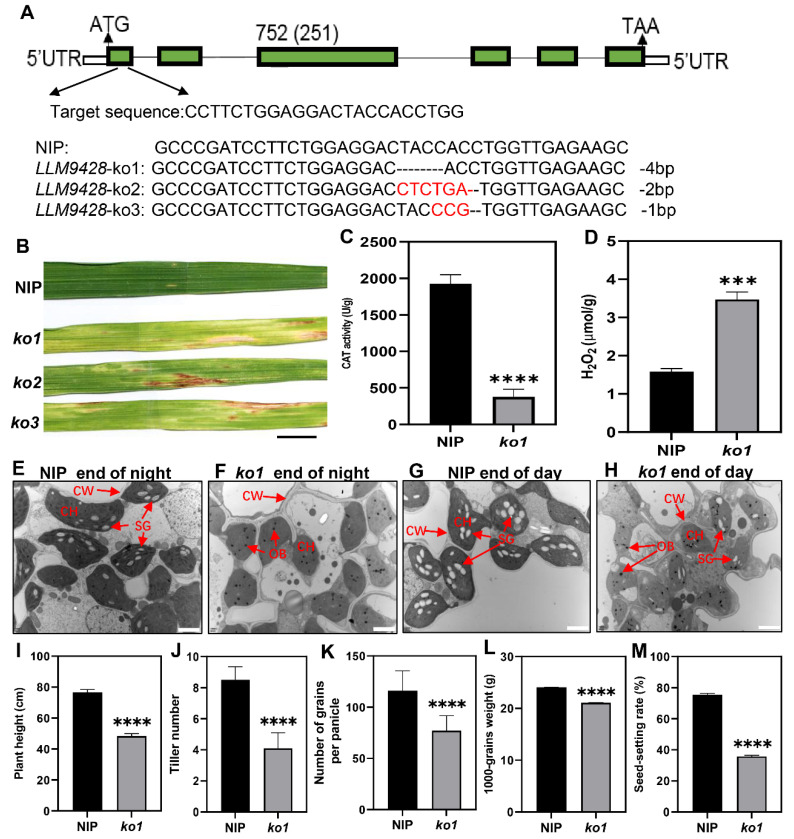
Phenotypic characterizations of knockout lines. (**A**) Deletion of nucleotides through CRISPR/cas9 in *ko-1*, *ko2* and *ko3*, (**B**) The phenotype of KO lines displaying lesion mimic spots on leaves at tillering stage. NIP: Nipponbare. Scale bar: 1 cm. (**C**) Changes of CAT activity in the leaf of NIP and *ko1* at the tillering stage, (**D**) Comparison of H_2_O_2_ content in the leaves between NIP and *ko1* at the tillering stage, (**E**) Sub-cellular structure of leaves in NIP at the end of the night, (**F**) Sub-cellular structure of leaves in *ko1* at the end of the night, (**G**) Sub-cellular structure of leaves in NIP at the end of the day, (**H**) Sub-cellular structure of leaves in *ko1* at the end of the day, where CW, cell wall, CH, chloroplast; SG, starch granules; OB, osmiophilic body. Scale bar: 2 μm in (**E**–**H**). Comparison of agronomic traits e.g plant height (**I**), tiller number (**J**), number of grains per panicle (**K**), 1000-grains weight (**L**), and seed setting rate (**M**). Data were obtained from 10 plants of NIP and *ko1*. Statistical analysis was performed using Student’s *t*-test, where *** indicates *p* < 0.001 in (**D**) and **** indicates *p* < 0.0001 in (**C**,**I**–**M**). Data presented in (**I**–**M**) were the average of *n* = 3 plants.

**Table 1 ijms-23-03827-t001:** Segregation ratios of F_2_ plants with and without LMM phenotype.

Population	Total Plants Observed	Plants Showing WT’s Phenotype	Plants Showing Mutant’s Phenotype	Chi-Square Test	Segregation Ratio
*llm9428* × Yixiang1B	584	452	132	χ^2^ = 1.66 < χ^2^_0.05,1_ = 3.84	3.42:1
Yixiang1B × *llm9428*	890	671	219	χ^2^ = 0.05 < χ^2^_0.05,1_ = 3.84	3.06:1

## Data Availability

The datasets supporting the conclusions of this article are included within the article.

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
