# Peer review of "Disruption of LLM9428/OsCATC Represses Starch Metabolism and Confers Enhanced Blast Resistance in Rice"

_ijms, 2022, doi:10.3390/ijms23073827_

Round 1
Reviewer 1 Report
The manuscript entitled "Disruption of LLM9428/OsCATC, encoding a CATALASE C, represses starch metabolism and confers enhanced blast resistance in rice" sets out to investigate the role of LLM9428/OsCATC in starch metabolism and resistance to rice blast. The llm9428 showed a stable phenotype of lesion mimic mutant, decreased cell viability and increased ROS, enhanced blast disease resistance, and Reduction in starch content and expression of its related genes. The study is on relevance and general interest to the journal's readers. However, I have several concerns about presenting the data that should be addressed before publication.
- The authors are highly recommended to avoid using a personal pronoun (e.g., We, our, etc.); they can use the third party in the past tense's passive voice.
- Any abbreviation must be associated with the full name at the first mention in the manuscript to allow the reader to follow up because not all the readers are familiar with the abbreviated terminology
- In the material and methods section, all the sources of cell lines, chemicals, and equipment need to be added or completed (add city, state, and country).
- The authors need to support the material and methods section with the appropriate citation, otherwise describe the method in detail.
- Figure 6 needs to be reproduced to be more clear, I recommend using color in A, B, and C
Reviewer 2 Report
The current manuscript “Disruption of LLM9428 OsCATC , encoding a CATALASE C, represses presses starch metabolism and confers enhanced blast resistance in rice” by Liao et al described the characterization and genetic analysis of the LLM mutant. The authors also described a gene (OsCATC) as a candidate gene and performed a genetic mapping and functional study. However, the manuscript has major flaws and therefore, should be rejected for publication in IJMS.
Following are the comments.
Title:
Please consider rephrasing the title and removing the commas (,) from it.
Line 19: Write full forms of abbreviation when first time use in the text
Check the grammar of the manuscript throughout. One example of misuse of "the” article is in line 44 “The leaf is the main source organ of photosynthesis in plants”.
The manuscript is lacking harmony. The authors did not keep the description of methods and results in proper order. Please write the results section according to the order you wrote the method section.
Line 60: Write full forms of abbreviation when first time use in the text
Where did the authors mention the mutant plants' development? How much EMS was applied? What rice variety was used? The whole procedure of mutant development is missing
Line 298-310: How old were the plants when samples were taken for the staining?
Line 315: Are > were
Line 319-321: Manufacturer details?
Line 334: Reference?
Line 336: Which blast? Mention the pathogen name and source.
Line 342: bacterial liquid? Could you please elaborate more? Please carefully re-write the inoculation process.
Line 348: check italic stay consistent in lines 352 and 358
Line 350-355: carefully re-read and rephrase
Line 371: why did the authors use japonica cv. Nipponbare for transgenic development?
Line 80-88: again missing all the information where the llm9428 came from? No details no references
What is the N in Fig C to G? What were mean and standard errors?
Line 116: check the spellings of pathogen name
Line 113-122: How old were the plants when inoculated?
Line 119: Check the uppercase with “Pathogenesis-related (PR)”
Line 120-122: No figure has been referred for results. Why only PR1a, PR1b, and PR10 were tested for the expression. Where are the other genes which were tested? Where are there primer details? Why not mentioned it in the method section properly?
Line 114-122: what was the lesion diameter? Why the authors did elaborate the figure 3 fully?
Line 152-153. Rephrase and refer to the figure.
Line 155 to 160: why do the authors check only “OsSSI, OsSSIIb, OsAGPL1 and OsAGPL3” where are the other genes which were tested? Where are there primer details? Why not mentioned it in the method section properly?
Line 171-200: The authors mentioned they developed two mapping populations and also BC1F1. The genetic analysis part is not clear. Make a clear table or figure for the populations developed and for what purpose they were used. How many individuals were developed and phenotype? What was the segregation ratio? All data should be clearly presented in a table form. Where is BC1F1?
Where is the Mutmap? The authors mentioned in method section line 351.
Line 178-179: Rephrase the sentence.
Where are the results of polymorphism of SSR markers? Where are the gel images of genotyping?
Line 180: How come suddenly a candidate gene appeared at chromosome 3? Where is the genetic linkage map? Where are recombinants?
Where are the sequences results of and evidence of SNP? Where is the nucleotide translation proof of amino acid flips?
The genetic analysis part is a total mess. The authors should write clearly again with evidence of results in case of re-submission.
Line 206-208: What do the authors want to say here?
Line 211: What is TEM?
The procedure of genome editing is also not elaborated properly. The authors should provide the proper evidence of genome-edited plants with gel image and deep sequencing results. After that phenotype change. These results are not enough to believe the successful editing and gene knockout.
Round 2
Reviewer 1 Report
The authors responded to all of my comments, so that, the manuscript could be published unless the editor has some other issues that were raised by the other reviewers.
Reviewer 2 Report
I appreciate the efforts made by the authors to improve the manuscript. The manuscript is good enough to be accepted for publication.